# Loss of Tsc1 in cerebellar Purkinje cells induces transcriptional and translation changes in FMRP target transcripts

Jasbir Singh Dalal[1†], Kellen Diamond Winden[1†], Catherine Lourdes Salussolia[1], Maria Sundberg[1], Achint Singh[1], Truc Thanh Pham[1], Pingzhu Zhou[2], William T Pu [2,3], Meghan T Miller[4], Mustafa Sahin[1,3]*

[1]Department of Neurology, Rosamund Stone Zander Translational Neuroscience Center, Kirby Neurobiology Center, Boston Children's Hospital, Boston, United States; [2]Department of Cardiology, Boston Children's Hospital, Boston, United States; [3]Harvard Medical School, Boston, United States; [4]Roche Pharma Research and Early Development, Neuroscience and Rare Diseases, Roche Innovation Center Basel, Basel, Switzerland

*For correspondence:
mustafa.sahin@childrens.harvard.edu

[†]These authors contributed equally to this work

**Abstract** Tuberous sclerosis complex (TSC) is a genetic disorder that is associated with multiple neurological manifestations. Previously, we demonstrated that *Tsc1* loss in cerebellar Purkinje cells (PCs) can cause altered social behavior in mice. Here, we performed detailed transcriptional and translational analyses of *Tsc1*-deficient PCs to understand the molecular alterations in these cells. We found that target transcripts of the Fragile X Mental Retardation Protein (FMRP) are reduced in mutant PCs with evidence of increased degradation. Surprisingly, we observed unchanged ribosomal binding for many of these genes using translating ribosome affinity purification. Finally, we found that multiple FMRP targets, including SHANK2, were reduced, suggesting that compensatory increases in ribosomal binding efficiency may be unable to overcome reduced transcript levels. These data further implicate dysfunction of FMRP and its targets in TSC and suggest that treatments aimed at restoring the function of these pathways may be beneficial.

## Introduction

Tuberous sclerosis complex (TSC) is a neurocutaneous disorder caused by germline heterozygous loss-of-function mutations in *TSC1* or *TSC2*. Among the neurological manifestations, approximately 50% of patients with TSC are diagnosed with autism spectrum disorder (ASD) (*Jeste et al., 2008*; *Jeste et al., 2016*; *Capal et al., 2017*). Disrupted development of the cerebellum is among a number of pathogenic processes that have been associated with ASD (*Fatemi et al., 2012*; *Becker and Stoodley, 2013*; *Rogers et al., 2013*), and cerebellar pathology has been identified in patients with TSC (*Weber et al., 2000*; *Eluvathingal et al., 2006*; *Vaughn et al., 2013*; *Weisenfeld et al., 2013*). We have previously demonstrated that loss of either one or both alleles of *Tsc1* in cerebellar Purkinje cells (PCs) results in abnormal social behavior (*Tsai et al., 2012*; *Tsai et al., 2018*). Therefore, we evaluated the molecular targets that are dysregulated in TSC1-deficient PCs to understand the mechanism by which loss of TSC1 leads to PC dysfunction and abnormal social behavior.

TSC1 and TSC2 form a protein complex (TSC1/2) that negatively regulates the mechanistic target of rapamycin (mTOR) pathway by inactivating RHEB. Both TSC1 and TSC2 are necessary for a functional TSC1/2 complex (*Hoogeveen-Westerveld et al., 2011*), and loss of either gene leads to the clinical manifestations of TSC that are indistinguishable from one another. Thus, loss of either TSC1 or TSC2 leads to unchecked activation of mTOR. mTOR is a kinase that regulates several cellular processes, including proliferation, protein synthesis, and autophagy (*Lipton and Sahin, 2014*;

*Liu and Sabatini, 2020*). The phosphorylation targets of mTOR depend on its surrounding adapter proteins, which define either the mTOR complex 1 (mTORC1) or mTOR complex 2 (mTORC2). mTOR signaling from either complex may affect transcriptional regulation of several genes and pathways, and we previously observed that loss of TSC2 in cortical neurons caused upregulation of the transcription factor ATF3 (*Nie et al., 2015*). However, mTORC1 also facilitates translation initiation and elongation through phosphorylation of 4EBP1/2 and ribosomal protein S6 (*Saxton and Sabatini, 2017*). Therefore, loss of TSC1/2 can cause complex changes in protein expression through modulation of transcription and translation in neurons.

In this study, we sought to characterize the effects of loss of *Tsc1* on PCs at both the level of transcription and translation. We performed RNA sequencing of total RNA from sorted PCs using FACS, and we used translating ribosome affinity purification (TRAP) to determine ribosomal binding of transcripts in PCs. By comparing these different levels of RNA regulation, we observed that many transcripts that bind to Fragile X Mental Retardation Protein (FMRP) maintain ribosomal binding despite reduced levels of transcripts overall. For one gene, *Shank2*, we found that the result of these opposing effects was reduced levels at PC synapses. These data highlight a complex interplay between the effects of mTOR activation of the regulation of transcript levels and ribosomal binding and provide further insight into the dysfunction of PCs in TSC.

## Results

### Loss of *Tsc1* in PCs leads to downregulation of gene expression

To assess the effect of loss of *Tsc1* on the transcriptome of PCs, we designed an experimental paradigm that allowed us to isolate PCs and measured their gene expression. We utilized our previously characterized animal model where *Tsc1* harbors loxP sites flanking exons 17 and 18, which leads to a null allele after Cre-mediated recombination (*Kwiatkowski et al., 2002*; *Meikle et al., 2007*). In this model, Cre is driven by the L7 (or PCP2) promoter that leads to expression of Cre specifically within PCs (*Tsai et al., 2012*). We then crossed these animals with an animal that expresses GFP under the same L7 promoter, leading to expression of GFP specifically within PCs (*Tomomura et al., 2001*). We performed immunostaining of control (*Tsc1*[+/+] L7GFP) and mutant (*Tsc1*[fl/fl] L7Cre;L7GFP) animals. We observed that PCs were specifically labeled with GFP and that loss of *Tsc1* led to increased soma size and phosphorylation of S6, consistent with mTORC1 activation (*Figure 1a,b*).

We then dissociated the cerebellum of control (N = 4) and mutant (N = 4) animals into single-cell suspensions and performed fluorescence-activated cell sorting (FACS) to specifically isolate GFP-positive PCs. After removal of debris, doublets, and dead cells, we found that GFP-positive PCs comprised approximately 7.7-10.8% of total events (*Figure 1—figure supplement 1*). We then isolated total RNA from these samples and performed RNA sequencing. Differential expression analysis between control and mutant animals identified 192 differentially expressed genes (FC > 2 and p-value<0.01; *Supplementary file 1*). Interestingly, most of the differentially expressed genes in this dataset were downregulated (72%) (*Figure 1c*). Indeed, we found that there was a bias toward downregulation in the overall gene expression, even among genes that did not show a significant change in gene expression (*Figure 1d*). This effect was more evident in genes with higher expression, and there was a statistically significant negative correlation between fold change and expression (r = –0.14, p<2.2e-16, Pearson correlation). These data demonstrate that loss of *Tsc1* results in a bias toward widespread transcriptional downregulation.

### Increased degradation of downregulated transcripts in *Tsc1*[fl/fl] PCs

We hypothesized that the bias toward reduced levels of a majority of transcripts may reflect a change in RNA stability caused by loss of *Tsc1*. Therefore, we asked whether there was a systematic alteration in the coverage of transcripts from the RNA sequencing data to understand whether RNA degradation pathways might be altered in mutant PCs. We used the aligned RNA sequencing data to calculate the coverage of each nucleotide across each transcript, and we focused on transcripts that showed an average coverage of at least one read per nucleotide. We then divided each transcript into 100 equally sized intervals, and we calculated the fraction of reads within each interval compared to the total number of reads across the transcript. When we compared the median coverage across all transcripts between mutant and control PCs, we observed that mutant PCs showed

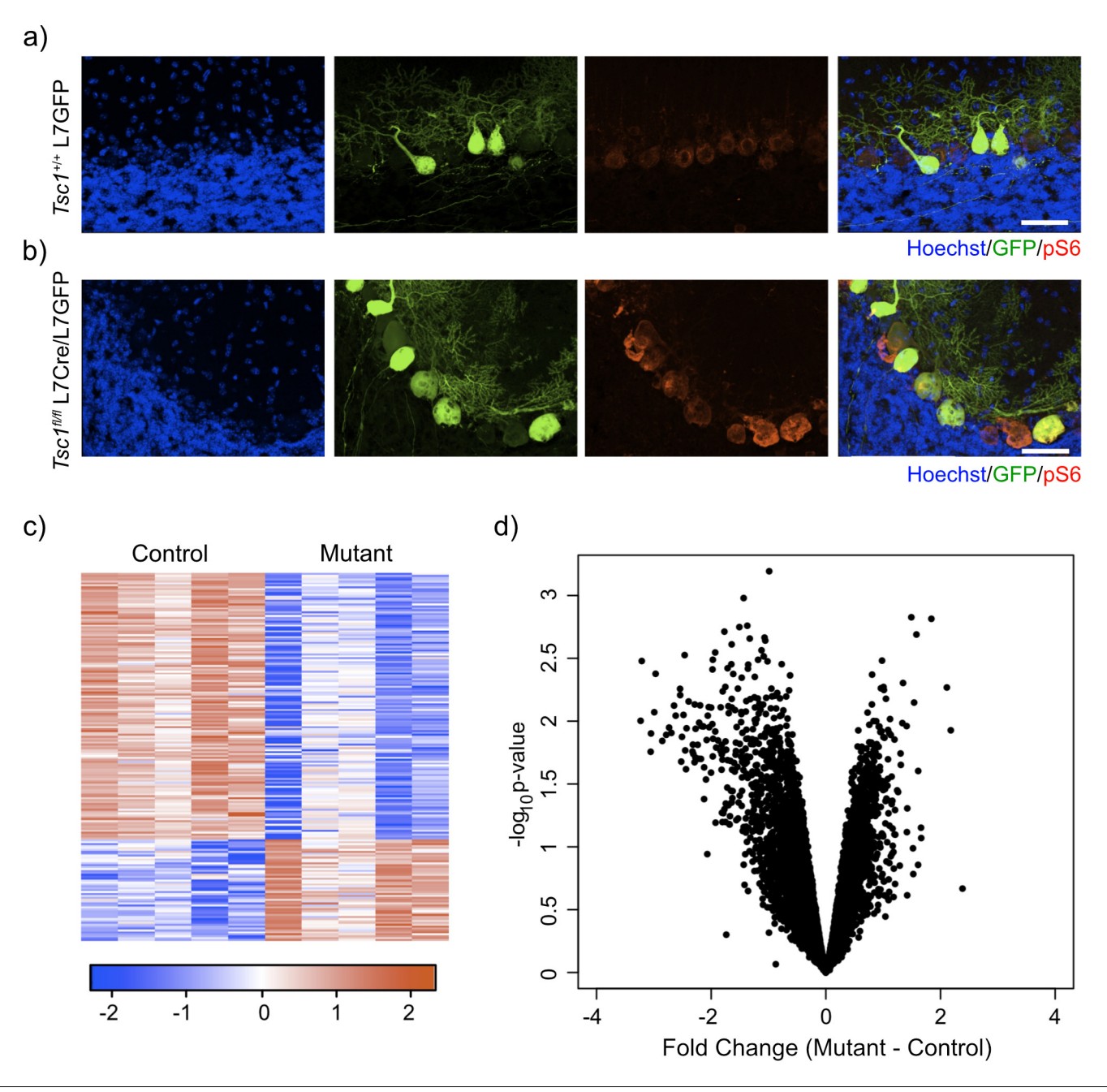

**Figure 1.** Labeling and sorting *Tsc1*fl/fl PCs show predominance of downregulation gene expression.  (a) Immunocytochemical characterization of wild-type control L7-GFP+ mouse PCs on P21 mouse cerebellum. Cell nuclei were stained with Hoechst. GFP+ PCs had dim expression of phosphorylated S6 (red). Scale bar 50 µm. (b) Immunocytochemical characterization of Tsc1fl/fl L7Cre+L7-GFP+ mouse PCs on P21 mouse cerebellum. Cell nuclei were stained with Hoechst. GFP+ PCs were strongly positive for pS6 (red). Scale bar 50 µm. (c) Heatmap of differentially expressed genes in mutant (*Tsc1*fl/fl; N = 4) vs control (*Tsc1*+/+; N = 4) PCs (FC > 2 and p-value<0.01; n = 679 genes). Each row of the heatmap represents the scaled expression of one gene, where red corresponds to higher relative expression and blue corresponds to lower relative expression. (d) This volcano plot shows the relationship between fold change, calculated Mutant-Control, and p-value across all genes. A majority of genes in this dataset demonstrate downregulation in the mutant PCs compared to control.

The online version of this article includes the following figure supplement(s) for figure 1:

**Figure supplement 1.** Gating strategy for FACS isolation of GFP labeled PCs.

reduced coverage at the 3' end of the transcript compared to control PCs (*Figure 2a*). These data show that there is a bias toward reduced coverage in the 3' region of transcripts in mutant PCs, suggesting that 3'–>5' RNA degradation pathways may be increased in mutant PCs.

To further understand the dependence between overall transcript expression and coverage across the transcript, we identified all significantly up- and downregulated transcripts and analyzed the difference in coverage of these genes between mutant and control across the 5' UTR, the coding sequence, and the 3' UTR to determine whether changes in one of these regions was responsible for driving the change in expression of the transcript. The 3' UTR showed significantly reduced coverage in the mutant compared to the control PCs among the downregulated genes (*Figure 2b*). However, there was no significant difference between the regions of the upregulated genes. These data suggest that there may be distinct mechanisms driving up- and downregulation of genes within TSC1-deficient PCs.

We then searched through the downregulated transcripts for those that showed greater changes in the 3' UTR than the overall transcript expression, identifying 978 transcripts (*Supplementary file 2*). We reasoned that this may be a core group of transcripts that are strongly affected by increased RNA degradation in mutant PCs. To understand what functions are associated with this group, we

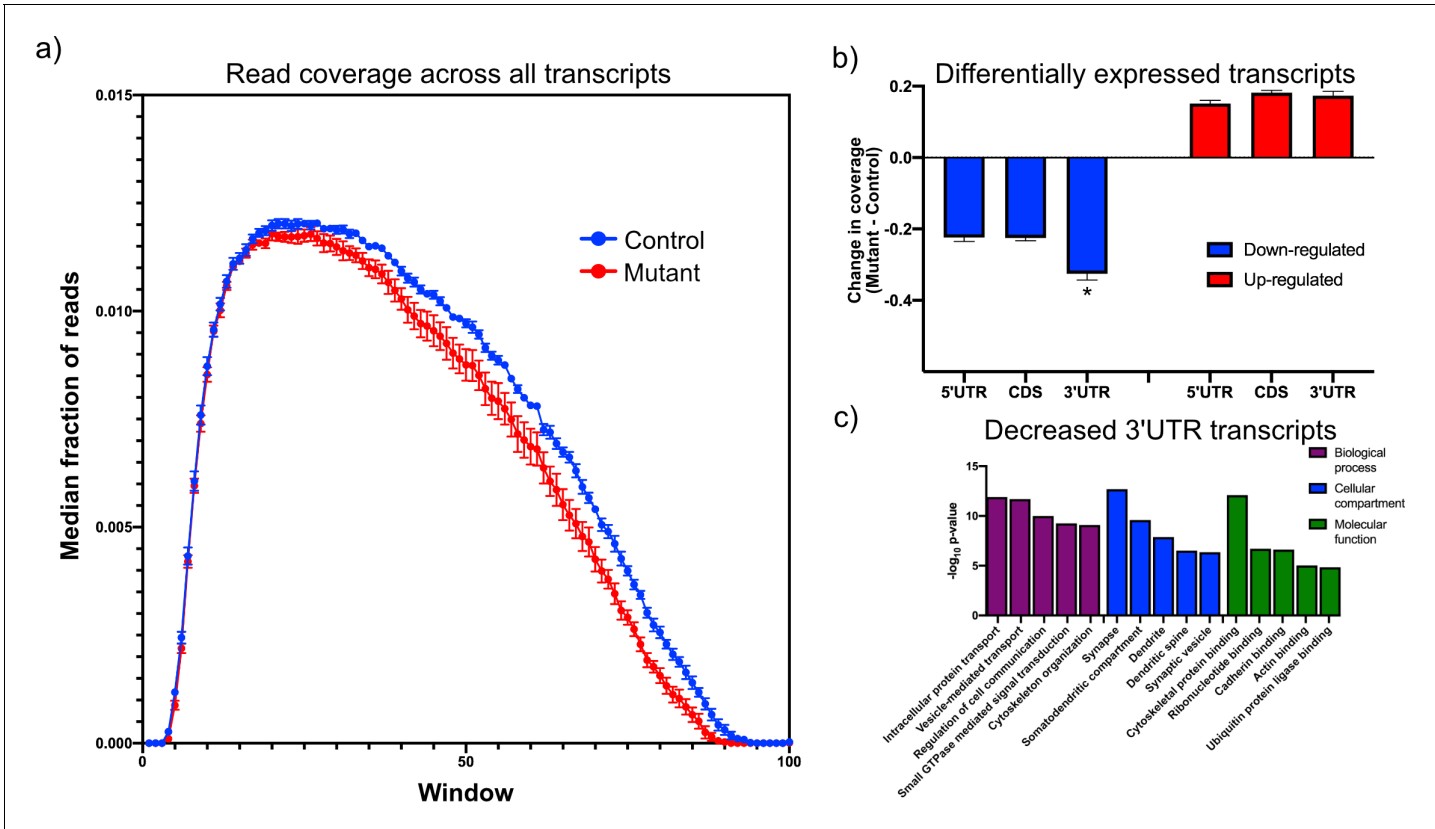

**Figure 2.** Downregulated transcripts in *Tsc1*-deficient PCs display enhanced reduction in the 3'UTR and are enriched in synaptic genes. (**a**) This plot shows the median coverage of reads across all downregulated transcripts. Each transcript was divided into 100 equally sized intervals and coverage was calculated across each of these intervals. The error bars show the SEM between five animals of each genotype, where mutant PCs are shown in red and control PCs are shown in blue. There is a bias for fewer reads across the 3' regions of transcripts in mutant PCs. (**b**) This bar plot shows the read coverage across different segments of each transcript for downregulated (blue) and upregulated (red) transcripts. Read coverage was calculated separately for the 5' UTR, coding sequence, and 3' UTR, and the error bars represent SEM. The 3' UTR of downregulated genes in mutant PCs demonstrate significantly reduced coverage compared to the 5' UTR and coding sequence (*p<0.001, ANOVA with Tukey's post hoc test). (**c**) This bar plot shows the enrichment of functional gene ontology categories among the downregulated transcripts with enhanced reduction in the 3' UTR. Several synapse-related categories are among the most enriched categories.

The online version of this article includes the following figure supplement(s) for figure 2:

**Figure supplement 1.** Characterization of transcripts with reduced 3'UTR expression.

performed gene ontology analysis and observed that these transcripts were strongly enriched in categories such as synapse, protein transport, and cytoskeletal binding (*Figure 2c*). In a prior study, we had observed that TSC2-deficient PCs that were derived from human-induced pluripotent stem cells showed downregulation of FMRP targets (*Sundberg et al., 2018*). Therefore, we examined this group of downregulated transcripts and found a highly significant enrichment of FMRP target genes (p=2.59e-55) (*Darnell et al., 2011*). Given that FMRP has been associated with longer genes (*Darnell et al., 2011*; *Ouwenga and Dougherty, 2015*), we examined the length of this group of transcripts. We observed that transcripts with reduced 3' UTR were significantly longer than a random set of transcripts with a similar distribution of expression (p=8.6e-5; Mann–Whitney U-test; *Figure 2—figure supplement 1*). We then examined this set of transcripts to determine whether they might be enriched in genes associated with neurological and psychiatric disorders, including ASD, intellectual disability, and schizophrenia. Interestingly, we only observed a significant enrichment in ASD risk genes (p=0.0048; hypergeometric probability), but not in the other disorders (*Figure 2—figure supplement 1*). Taken together, these data demonstrate that downregulated genes with alterations in the 3' UTR are FMRP targets with increased transcript length and are associated with synaptic functions and ASD susceptibility genes. Therefore, perturbation of RNA decay pathways in mutant PCs may substantially affect PC physiology.

## TRAP of *Tsc1*<sup>fl/fl</sup> PCs does not reflect decreased total transcript levels

mTOR also exerts strong effects at the level of translation, and therefore, we decided to evaluate ribosomal binding of transcripts within PCs to determine whether changes in translation mechanisms could compensate for or exacerbate the alterations that we had observed in the transcriptome. The TRAP paradigm utilizes a labeled ribosomal subunit (RPL10A-EGFP) that can be expressed in a subset of cells to specifically profile ribosomal binding of transcripts (*Heiman et al., 2014*). For these studies, we used animals that have a floxed-STOP RPL10A-EGFP in the constitutively active Rosa26 locus, which leads to EGFP-labeled ribosomes after Cre-mediated recombination (*Figure 3a*). We crossed these animals to the *Tsc1*/L7Cre animals described above. We then performed immunostaining of these animals, and we found that GFP was expressed specifically within calbindin-positive neurons, which is a known marker of PCs (*Figure 3b*).

We performed RNA sequencing on TRAP samples from mutant (N = 3) and control PCs (N = 3). We then focused on genes that showed significant changes in ribosomal binding between mutant and control PCs. This analysis identified 165 differentially bound genes (FC > 2 and p-value<0.01; *Supplementary file 3*). Interestingly, nearly equal numbers of genes showed increased and decreased ribosomal binding compared to control (*Figure 3c*). Consistent with these data, we found that there was no overall bias toward downregulation of ribosome bound mRNAs (*Figure 3d*), in contrast to the total RNA data obtained from sorted PCs.

## Translation efficiency is increased in *Tsc1*<sup>fl/fl</sup> PCs

mTOR is an important regulator of multiple aspects of translation, including initiation, elongation, and ribogenesis. Therefore, we examined ribosomal loading by calculating the translation efficiency (TE) for each gene, which is the difference between the ribosomal bound level of a gene and its total abundance in the cell (*Thoreen et al., 2012*). We then calculated the change in TE ($\Delta$TE) between mutant and control animals for each gene. To estimate the variability of TE and determine whether changes in TE were statistically significant, we calculated $\Delta$TE on subsets of the data and calculating the Z-score of each distribution. We found 1808 genes that showed a $\Delta$TE Z-score greater than 4, which corresponded to an uncorrected p-value<3.2e-5 (*Supplementary file 4*). A majority of these genes (1131) showed an increase in $\Delta$TE in mutant PCs compared to control (*Figure 4a*). These data demonstrate that activation of mTOR through loss of TSC in PCs leads to increased TE, consistent with known roles of the mTOR pathway in facilitating protein synthesis.

Known downstream targets of mTOR impact several aspects of protein synthesis in general, but studies have also found that mTOR can influence translation of specific transcripts. One specific motif that has been shown to increase translation in an mTOR related manner is the 5' terminal oligopyrimidine (5' TOP) motif, which is a pyrimidine rich motif near the transcription start site that is thought to facilitate initiation (*Thoreen et al., 2012*). In addition, reduction of FMRP has also been shown to potentiate ribosomal binding of TOP genes (*Das Sharma et al., 2019*). Therefore, we

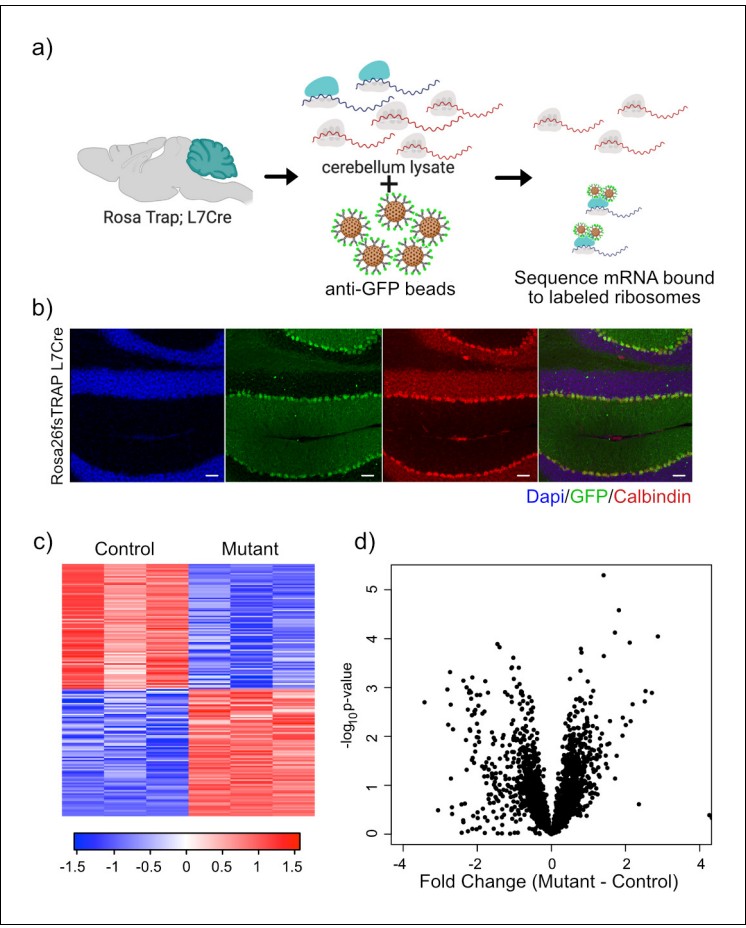

**Figure 3.** Expression of GFP-tagged ribosomes and ribosomal binding of transcripts in PCs.  (a) Schematic of TRAP protocol for identifying ribosomally bound transcripts specifically within PCs. (b) Anti-GFP immunofluorescence of in Rosa26fsTRAP L7Cre animals shows labeling of the PCs of the cerebellum. Sagittal sections from P56 Rosa-Trap animals expressing Cre under Pcp2 promoter (L7Cre) were stained with GFP (green), calbindin (red), and DAPI (blue). Scale bars, 10 μm. (c) Heatmap of TRAP levels of genes that were previously identified to be downregulated at the transcript level. Each row of the heatmap represents the scaled expression of one transcript, where red corresponds to higher relative expression and blue corresponds to lower relative expression. (d) This histogram shows the fold changes of the TRAP values for the previously identified downregulated transcripts between mutant and control PCs. Despite being downregulated at the total mRNA level, these transcripts generally show maintained ribosomal binding.

searched for 5' TOP motifs in the group of genes in mutant PCs that show increased TE. Although known TOP genes from one study (*Thoreen et al., 2012*) did not demonstrate a significant increase in TE, we found that all genes with a predicted TOP motif (*Yamashita et al., 2008*) displayed a significantly greater ΔTE compared to all other genes (p=0.004, T-test; *Figure 4b*). Consistent with this increase in TE, we found that these predicted TOP genes showed an overall positive fold change in ribosomal binding without a clear shift in total RNA abundance (*Figure 4c*). Other studies have found that length and/or complexity in the 5' UTR region was an important feature of mTOR-regulated genes (*Gandin et al., 2016*). However, we found that there was no increase in the length of the 5' UTR of transcripts with increased ΔTE (*Figure 4d*). We then examined the downregulated genes with exaggerated reduction of the 3' UTR identified above, and we found a remarkably significant overlap between these genes and those with significantly increased ΔTE (*Figure 4e*; p=2.08e-133; Hypergeometric probability). Interestingly, as opposed to the predicted TOP genes, these genes display downregulation at the total transcript level and unchanged ribosomal binding, leading to their increased TE (*Figure 4f*). These data indicate that genes with enhanced downregulation of

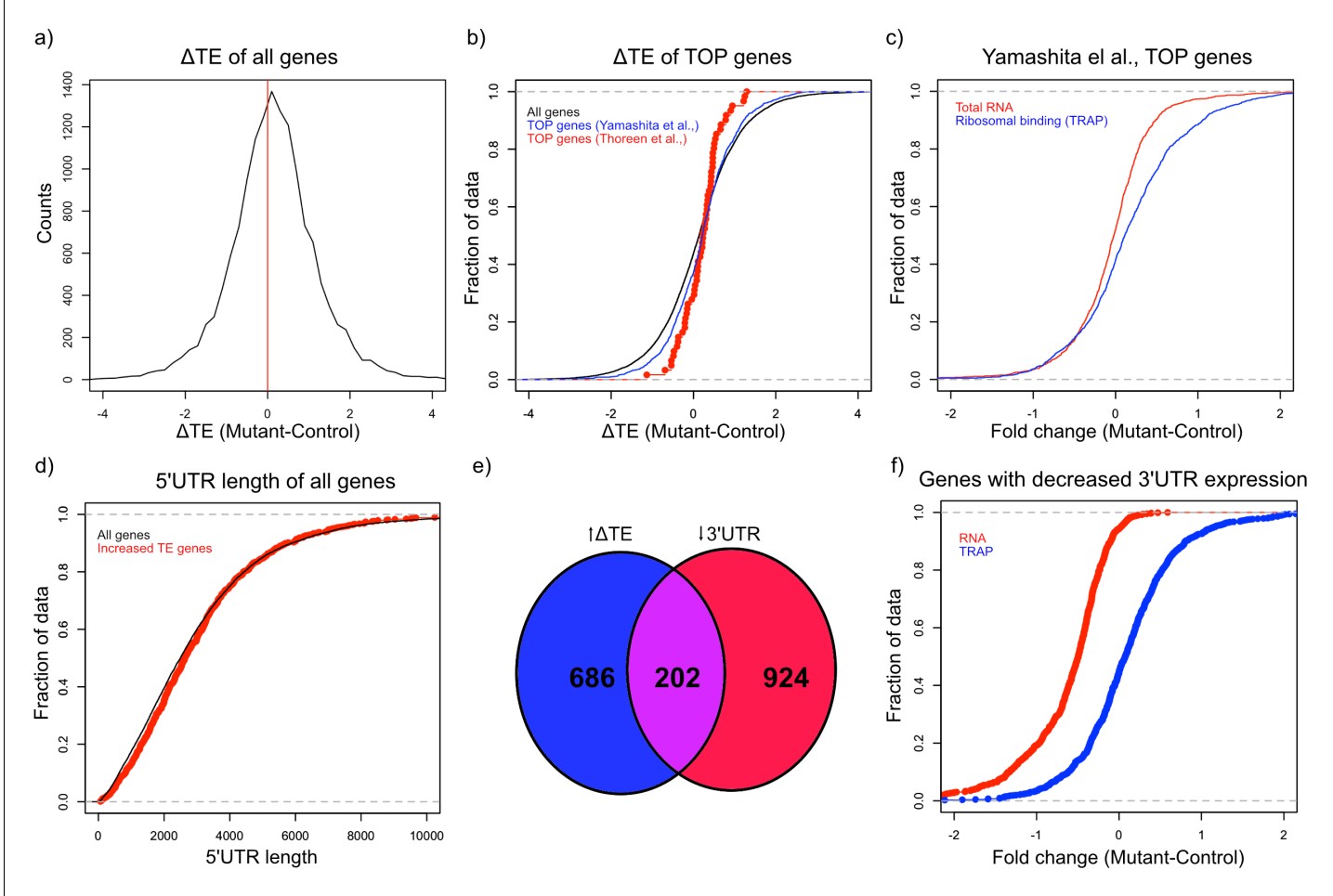

**Figure 4.** Alterations in TE in *Tsc1*<sup>fl/fl</sup> PCs. (a) This histogram shows the ΔTE between mutant and control PCs, and the vertical red line in the middle identifies the point where ΔTE = 0. A majority of the genes are situated to the right of the red line demonstrating increased ΔTE in mutant PCs. (b) These empiric cumulative distribution function (ECDF) plots show the distribution for all genes (black), known TOP genes from Thoreen et al. (red), and predicted TOP genes from Yamashita et al. (blue). There is a slight shift towards higher ΔTE for Yamashita et al. predicted TOP genes compared to all genes (p=0.004, T-test). (c) ECDF plot of fold changes for the total RNA and TRAP data for the predicted TOP genes from Yamashita et al., demonstrating that these genes have a slightly higher ribosomal binding in CC PCs without a change in the total RNA levels. (d) ECDF plot of the 5' UTR length for all genes (black) and genes with significantly increased ΔTE (red). There is no difference between these two groups of genes (p>0.05, Mann–Whitney U-test). (e) Venn diagram showing the highly significant overlap between genes with significantly increased ΔTE and downregulated transcripts with reduced coverage in the 3' UTR. (f) ECDF plot of fold changes for the total RNA and TRAP data for downregulated transcripts with reduced 3' UTR coverage. These data show that the increased ΔTE of these genes is due to reduced total RNA levels with maintained ribosomal binding levels.

the 3' UTR, which are enriched in FMRP targets, show increased relative ribosomal binding with loss of TSC1.

## Increased TE of FMRP targets in *Tsc1*<sup>fl/fl</sup> PCs

Given our observations that FMRP targets tended to be downregulated at the transcript level but maintain their ribosomal binding, we decided to more carefully examine FMRP targets in TSC1-deficient PCs. We identified a group of 150 FMRP target genes based on known FMRP binding in prior studies (*Darnell et al., 2011*) and decreased expression in *Tsc1*<sup>fl/fl</sup> L7Cre PCs, and we re-assessed their transcript level expression and RNA binding using amplicon-based quantification. We created amplicon-based libraries from total RNA from FACS PCs and TRAP RNA, so that these data would be directly comparable without technical confounding due to different library preparation methods. In addition, the use of amplicons allowed interrogation of the same region of the transcript in both

total and TRAP RNA. Consistent with our prior data, we confirmed that most of the FMRP targets were downregulated in mutant PCs compared to control PCs (*Figure 5a*). However, these same genes showed no significant change in overall distribution in their ribosomal binding (*Figure 5b*). Therefore, these data confirm our observation of differential effects of loss of *Tsc1* on FMRP targets at the transcript and ribosome-bound levels.

## Changes in protein expression in *Tsc1*<sup>fl/fl</sup> PCs

Previously, we had observed an increase in translation efficiency in predicted 5′ TOP genes, and therefore, we decided to investigate the protein levels of these genes. We examined the expression of predicted 5′ TOP genes with increased TE in the cerebellum using the Allen Brain Atlas, and we identified two potential candidates, *Cdk16* and *Rpl28*, that appeared to be enriched in PCs. We then assayed the expression of these genes using immunoblotting on control and mutant whole cerebellum. Interestingly, we did not observe any significant changes in their expression at the protein level (*Figure 6—figure supplement 1*).

We then investigated the resulting protein levels for several FMRP target genes because of the discrepant findings between gene expression and ribosomal binding. We carefully examined known FMRP targets with reduced mRNA expression and unchanged ribosomal binding for specific expression within PCs using the Allen Brain Atlas. We identified three targets to test for changes at the protein level, including *Dlg3*, *Kif3c*, and *Shank2*. We performed immunoblots of *Tsc1*<sup>fl/fl</sup> and *Tsc1*<sup>fl/fl</sup> L7Cre cerebellum at P42–P45 (N = 6 animals of each genotype), and we observed a significant reduction of DLG3 and KIF3C at the protein level in mutant cerebellum (*Figure 6a,b*). We then performed immunohistochemistry of SHANK2 in the cerebellum of *Tsc1*<sup>+/+</sup> and *Tsc1*<sup>fl/fl</sup> L7Cre;L7GFP animals (N = 4 animals of each genotype). We used confocal microscopy to identify GFP-positive PC dendrites in the molecular layer, and measured SHANK2+ punctae within these processes. Interestingly, we observed a significant decrease in the intensity of SHANK2+ punctae in mutant PCs compared to control (*Figure 6c–e*). These data suggest that the increase ribosomal binding efficiency of

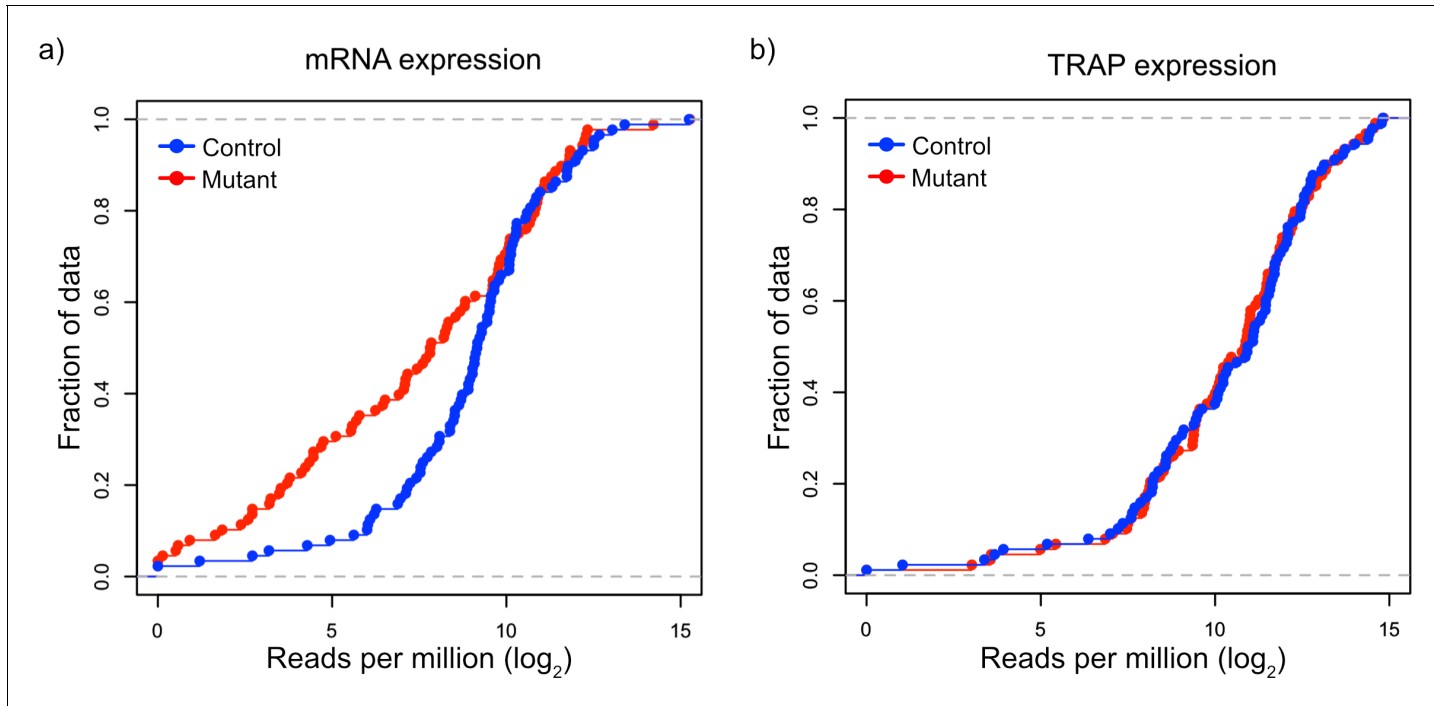

**Figure 5.** Validation of downregulation of transcripts and maintained ribosomal loading in *Tsc1*-deficient PCs. (**a**) This cumulative distribution function shows the distribution of total mRNA expression from sorted PCs for 150 FMRP target genes, previously found to be downregulated in mutant PCs, between mutant (red) and control (blue) PCs. (**b**) This cumulative distribution function shows the TRAP values for the same 150 genes between mutant (red) and control (blue) PCs. Despite the obvious downregulation of many genes at the total mRNA level, the ribosomal loading of these transcripts appears mostly unchanged.

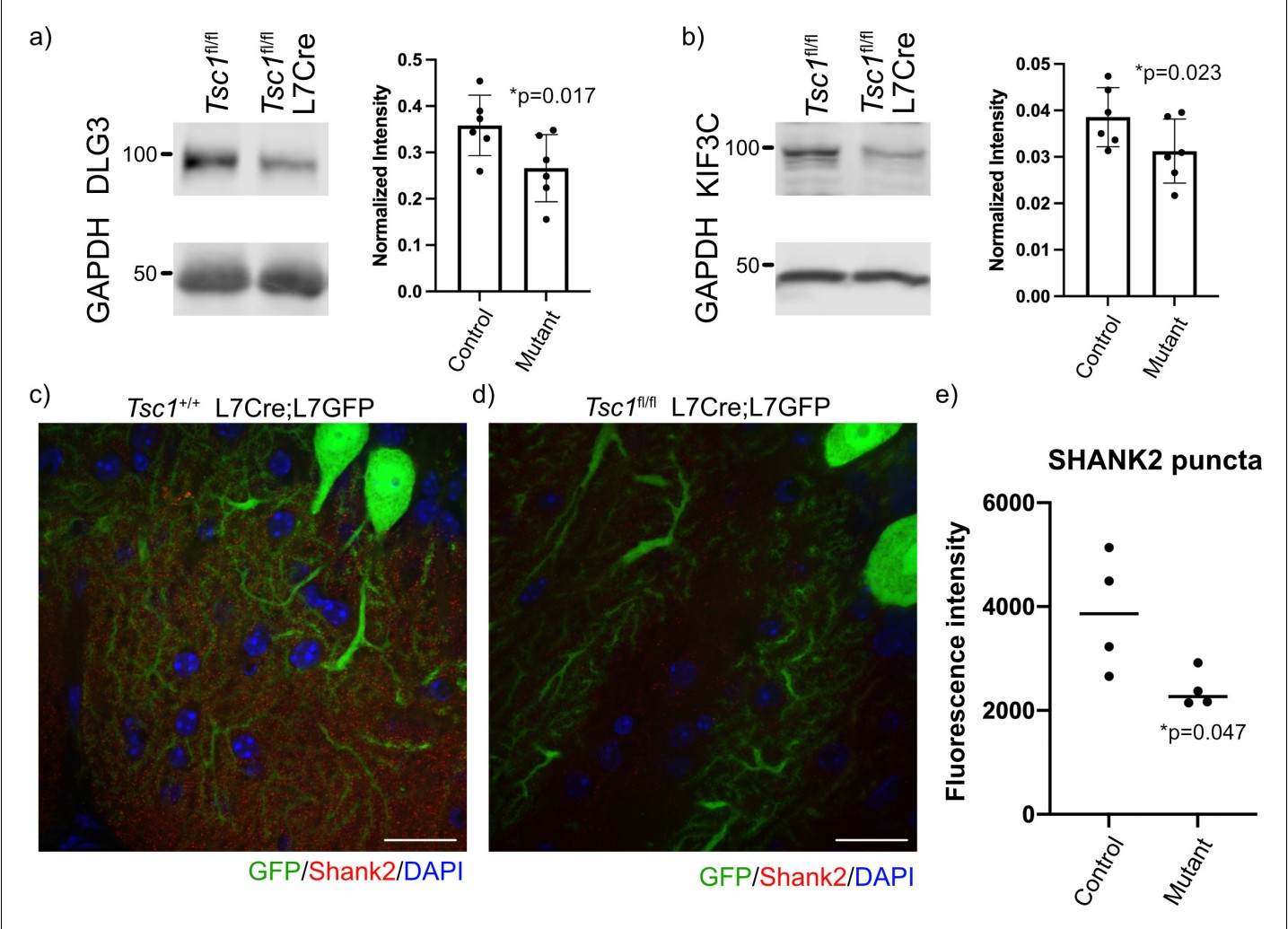

**Figure 6.** Reduced expression of FMRP targets in mutant cerebellum. Protein expression of (**a**) DLG3 and (**b**) KIF3C assessed by western blot of whole cerebellum lysates from control (*Tsc1*[fl/fl]) and mutant (*Tsc1*[fl/fl] L7Cre) animals (N = 6). Bands were quantified using densitometry, and p-values calculated using a paired T-test, where littermates were paired together. Confocal images of staining of SHANK2 in (**c**) *Tsc1*[+/+] L7Cre/L7GFP and (**d**) *Tsc1*[fl/fl] L7Cre/L7GFP animals. SHANK2 is present in punctae along PC dendrites in the molecular layer, and GFP-labeled PCs are shown in green. Scale bars = 20 μm. (**e**) Bar plot showing the quantification of the intensity of SHANK2+ punctae in control and mutant animals. Each data point represents the mean value across 10 images from three–four independent slices from the same animal. N = 4 animals. p=0.047 (T-test).

The online version of this article includes the following figure supplement(s) for figure 6:

**Figure supplement 1.** Western blot controls and 5' TOP genes.

these transcripts in *Tsc1*-deficient PCs is unable to overcome their overall down regulation, resulting in a decreased level of the protein.

## Discussion

The cerebellum and PCs, in particular, have been implicated in the development of abnormal social behavior in TSC (*Tsai et al., 2012*; *Tsai et al., 2018*). Therefore, we have endeavored to understand the molecular abnormalities present in TSC1-deficient PCs. We observed that the majority of differentially expressed genes between mutant and control PCs were downregulated. We also found that these downregulated genes in mutant PCs were enriched in FMRP targets and showed reduced coverage at the 3' region, suggestive of increased RNA decay. In contrast, we did not observe a reduced ribosomal binding of these genes, implicating processes that facilitate more efficient loading of these transcripts onto ribosomes. Finally, we observed that several FMRP targets displayed

reduced protein expression, implying that compensatory increases in ribosomal binding efficiency may be insufficient to overcome overall reduced transcript levels. These data suggest that dysregulation of FMRP target genes at the transcript level, rather than the ribosomal binding level, may drive molecular dysregulation in *Tsc1* mutant PCs and underlie the development of synaptic abnormalities in these neurons.

We observed that there was a notable bias toward down-regulation of genes in mutant PCs, and we found that this was driven by exaggerated reduction of the coverage of the 3′ region, especially the 3′ UTR. Prior studies have reported that mTOR can affect RNA stability, but most of these studies showed that treatment with rapamycin reduced stability of certain transcripts (*Banholzer et al., 1997*; *Hashemolhosseini et al., 1998*; *Albig and Decker, 2001*). Rapamycin reduced the stability of specific transcripts in yeast through increasing the rate of deadenylation, although the mediator of this effect was not identified (*Albig and Decker, 2001*). Inhibition of mTORC1 with rapamycin was also shown to facilitate RNA degradation through the nonsense mediated decay pathway, but this effect was shown to be associated with alterations in translation machinery associated with the 5′ cap (*Martinez-Nunez et al., 2017*). It is likely that our observations contradict these earlier studies because none of the prior studies assayed hyperactive mTORC1 or were performed in neurons, and the downstream machinery may be substantially different in mutant PCs. Canonically, RNA degradation is preceded by deadenylation of the transcript, rendering the molecule susceptible to exonuclease activity from either the 5′ or 3′ end (*Łabno et al., 2016*). We hypothesize that 3′→5′ decay activity is increased in mutant PCs because of the reduction in coverage in the 3′ region of transcripts. The best characterized 3′→5′ exonucleases are *Dis3*, *Dis3L*, and *Dis3L2*, which differ in their association with the RNA exosome complex (*Łabno et al., 2016*; *Dos Santos et al., 2018*). Interestingly, 20 of 24 genes associated with the exosome show modest but non-significant increases in expression in *Tsc1*<sup>fl/fl</sup> PCs in this dataset (*Supplementary file 5*). DIS3 was also found to be upregulated due to ER stress in *Caenorhabditis elegans* (*Sakaki et al., 2012*), and we have found that TSC2-deficient neurons show evidence of ER stress (*Di Nardo et al., 2009*). In addition, the RNA exosome complex is involved in ribosomal RNA maturation (*Kobyłecki et al., 2018*; *Pirouz et al., 2019*), and mTORC1 is known to increase ribosomal biogenesis (*Gentilella et al., 2015*). Therefore, it is possible that mTORC1 may induce the RNA exosome by stimulating the production of ribosomes, and the RNA decay phenomenon that we observe could be a side effect of this process.

Another striking finding of this study is the maintenance of ribosomal binding of transcripts despite down-regulation of the gene at the transcript level. These data suggest that the primary abnormality in mutant PCs may be at the transcript level, while the increase in relative ribosomal binding is a compensatory effect. Interestingly, the genes that show this pattern of reduced transcript level with maintained ribosomal binding are highly enriched in targets of FMRP. We also observed a similar decreased expression of FMRP target transcripts in iPSC-derived PCs with mutations in *TSC2* (*Sundberg et al., 2018*), corroborating the findings of this study. FMRP is an RNA binding protein that is lost in Fragile X Syndrome, which is characterized by high rates of intellectual disability and ASD (*Niu et al., 2017*). FMRP is known to be a repressor of translation (*Feng et al., 1997*; *Brown et al., 2001*), but its effects on the ribosomal binding of specific transcripts are less clear (*Thomson et al., 2017*). In addition, FMRP can bind to ribosomes and reduce their ability to synthesize new proteins, suggesting a more general effect on translation (*Chen et al., 2014*). Therefore, it is possible that dysfunction in FMRP could facilitate binding of specific transcripts to ribosome despite their reduced abundance at the total transcript level. FMRP has also been shown to play a role in mRNA stability (*Zalfa et al., 2007*; *De Rubeis and Bagni, 2010*), which suggests that it could also contribute to the increased RNA degradation and downregulation of its target transcripts. A recent study demonstrated that loss of FMRP can preferentially destabilize transcripts in neurons based on their codon optimality (*Shu et al., 2020*). In contrast, another recent study that examined hippocampal slices from *Tsc2*<sup>+/−</sup> mice reported that the expression of FMRP targets was increased and their ribosomal binding was decreased (*Hien et al., 2020*). It is possible, and in fact likely, that there are cell type differences in the functions of TSC1/TSC2 and FMRP in the brain leading to the discrepancy between this prior study and our data. Our study further reinforces the connection between mTOR and FMRP and highlights the role of mTOR hyperactivation in altering both the total abundance and ribosomal binding of FMRP target genes.

Given the varied effects on gene expression and ribosomal binding, there are likely to be pleiotropic effects on *Tsc1*-deficient PCs. We did observe a subtle but significant increase in predicted 5′

TOP genes, which is consistent with studies that have demonstrated that both mTOR activation and loss of FMRP can facilitate ribosomal binding of these mRNAs (*Thoreen et al., 2012*; *Das Sharma et al., 2019*). However, we were not able to observe these changes at the protein level. Given that these genes are involved in general cellular processes, it is possible that other cell types in the cerebellum contribute to their expression, diluting any change in expression in PCs. Therefore, further studies using more sensitive and specific techniques will be required to determine the consequences of increased ribosomal binding of 5' TOP genes in *Tsc1*-deficient PCs. In contrast, we observe reduced protein levels of several FMRP targets. *Kif3c* is an anterograde motor that is implicated in both axonal and dendritic trafficking (*Davidovic et al., 2007*; *Gumy et al., 2013*). In addition, *Dlg3*, which encodes for post-synaptic density protein 93, and *Shank2* are scaffolding proteins present at the post-synaptic density and highly expressed in PCs (*McGee et al., 2001*; *Eltokhi et al., 2018*). *Dlg3* is implicated in X-linked intellectual disability, while *Shank2* is a known risk gene for ASD (*Tarpey et al., 2004*; *Satterstrom et al., 2020*); therefore, dysregulation of these genes may contribute to neurological and behavioral abnormalities. Deletion of *Shank2* in PCs has been shown to alter levels of glutamate receptors in dendritic spines and lead to abnormal social behavior in these animals (*Peter et al., 2016*). Another study that deleted *Shank2* in PCs found alterations in excitation onto PCs but did not observe abnormal social behavior (*Ha et al., 2016*). These data suggest that proper regulation of *Shank2* in PCs is critical for normal function of cerebellar circuits. We found that the apparently compensated ribosomal binding each of these FMRP targets is insufficient to overcome its downregulation at the transcript level, leading to its downregulation at the protein level. Therefore, these data suggest that reduced expression of these synaptic proteins may contribute to the development of altered cerebellar circuitry and abnormal social behavior.

In this study, we performed profiling of the total transcriptome and ribosomally bound transcripts in PCs with a deletion of *Tsc1*. We observed changes at the transcript level that suggest alterations in RNA degradation and compensatory changes at the ribosome level. These data indicate the presence of novel mechanisms underlying PC dysfunction in TSC, which may be associated with the development of abnormal social behavior. Further study of these pathways may provide new avenues for therapeutics for ASD in TSC and related neurodevelopmental disorders.

# Materials and methods

## Key resources table

| Reagent type (species) or resource | Designation | Source or reference | Identifiers | Additional information |
|---|---|---|---|---|
| Strain, strain background (*Mus musculus*) | *Tsc1*^fl/fl | Jackson Labs | (Stock# 005680) | |
| Strain, strain background (*Mus musculus*) | Rosa26fsTRAP | Jackson Labs | (Stock# 022367) | |
| Strain, strain background (*Mus musculus*) | L7Cre | Jackson Labs | (Stock# 010536) | |
| Strain, strain background (*Mus musculus*) | L7GFP | Jackson Labs | (Stock# 004690) | |
| Antibody | Anti-GFP (Mouse monoclonal) | Memorial Sloan Kettering Centre | HtzGFP-19F7 and HtzGFP-19C8 | 50 µg per IP |
| Antibody | Anti-DLG3 (PSD93) (Mouse monoclonal) | Biolegend | 818302 | (1:1000) |
| Antibody | Anti-KIF3C (Rabbit polyclonal) | Proteintech | 144333–1-AP | (1:800) |
| Antibody | Anti-CDK16 (PCTAIRE) (Rabbit polyclonal) | Proteintech | 10102–1-AP | (1:750) |
| Antibody | Anti-Rpl28 (Rabbit polyclonal) | Proteintech | 16649–1-AP | (1:1000) |
| Antibody | Anti-Gapdh (Mouse monoclonal) | ThermoFisher | AM4300 | (1:10,000) |

*Continued on next page*

*Continued*

| Reagent type (species) or resource | Designation | Source or reference | Identifiers | Additional information |
|---|---|---|---|---|
| Antibody | Anti-B Actin (Mouse Monoclonal) | Cell Signaling | 3700 | (1:5000) |
| Antibody | Anti-GFP (Chicken polyclonal) | ThermoFisher | A10262 | (1:1000) |
| Antibody | Anti-Shank2 (Rabbit polyclonal) | ThermoFisher | PA5-78652 | (1:500) |
| Peptide, recombinant protein | Purified Recombinant Biotinylated Protein L | Pierce | 29997 | |
| Chemical compound, drug | Streptavidin MyOne T1 Dynabeads | ThermoFisher | 65601 | |
| Chemical compound, drug | Cycloheximide | Sigma Aldrich | 01810–1G | |
| Chemical compound, drug | RNasin | Promega | N2515 | |
| Chemical compound, drug | Superasin | ThermoFisher | AM2694 | |
| Chemical compound, drug | DHPC | Avanti Polar Lipids | 850306 | |
| Chemical compound, drug | TRIzol | ThermoFisher | 15596026 | |
| Chemical compound, drug | Papain | Worthington | LS003126 | |
| Chemical compound, drug | DNase I | Worthington | LK003172 | |
| Commercial assay or kit | RNAeasy Minelute kit | Qiagen | 74204 | |
| Software, algorithm | R v4.0.2 | https://cran.r-project.org/ | | |
| Software, algorithm | Prism v9 | GraphPad | | |
| Software, algorithm | Python v2.7 | https://www.python.org/ | | |

## Mice

All animal procedures were carried out in accordance with the Guide for the Humane Use and Care of Laboratory Animals, and all procedures in this study were approved by the Animal Care and Use Committee of Boston Children's Hospital. All animals were kept in ARCH animal house facility under 12 hr light/dark cycle with food and water available ad libitum. $Tsc1^{fl/fl}$ mice possess loxP sites flanking exons 17 and 18 of the $Tsc1$ gene (stock# 005680) and were obtained from Jackson labs (*Kwiatkowski et al., 2002*). The Rosa26fsTRAP mouse line (stock# 022367) and the L7Cre mouse line (stock# 010536) were obtained from Jackson labs. The L7GFP line was also obtained from Jackson labs (stock# 004690). The L7GFP mice were on a C57BL/6J background and were bred with $Tsc1^{fl/+}$ L7Cre animals to generate $Tsc1^{fl/+}$ L7Cre;L7GFP animals, which were bred together to generate $Tsc1^{fl/fl}$ or $Tsc1^{+/+}$ L7Cre;L7GFP animals. The Rosa26fsTRAP mice were on a C57BL/6 j background and were bred with $Tsc1^{fl/+}$ L7Cre animals to generate $Tsc1^{fl/+}$ L7Cre;Rosa26fsTRAP animals. These animals were bred together to generate either $Tsc1^{fl/fl}$ or $Tsc1^{+/+}$ L7Cre;Rosa26fsTRAP animals.

## Fluorescence-activated cell sorting

For isolation of PCs, we used $Tsc1^{+/+}$ or $Tsc1^{fl/fl}$ L7Cre/L7GFP animals. The cerebellum was microdissected from either P21 or P42 animals and chopped into small pieces before incubation with papain digestion buffer (1× HBSS medium containing 40 U/ml Papain [Worthington Biochemical, cat # LS003126], DNase I 100 units/ml [Worthington Biochemical, cat # LK003172], and 30 mM glucose)

for 30–45 min at 37°C with 5% $CO_2$. Tissue was dissociated into single cells by trituration with a fire-polished glass pipette. Cell suspensions were washed twice with papain wash buffer (1× HBSS supplemented 30 mM glucose, and 10% FBS and 1× penicillin/streptomycin antibiotic cocktail). Cells were then resuspended in FACS buffer (1× HBSS with 1× pen/strep solution and 2% BSA [sigma, cat # A9576]) and filtered through 40 µM cell strainer (Falcon). Cells were centrifuged 200 × g for 3 min and suspended into FACS buffer (1× HBSS and 1% BSA). Cells were sorted with FACS-Aria II (BD Bioscience) using 100 µM nozzle setting. Cells were identified from debris based on the forward and side scatter, and doublets were removed using FSC-H and SSC-W/SSC-H. Cells that were GFP positive and Propidium Iodide (PI) negative were collected into FACS buffer, and between 1 and 2 million cells were collected per animal. Cells were washed once with PBS and total RNA isolated using Qiagen mini Plus kit according to manufacturer's instructions (Qiagen). RNA quality and integrity were assessed by analysis in Bioanalyzer. RNA samples with RIN values > 8 were used for library preparation according to manufacturer's instructions (SmartSeqv4 RNASeq kit, Clontech) or manual library preparation for Ampliseq analysis (Ion AmpliSeq Lib Kit Plus). From each mouse, an average of 1,500–1,000,000 GFP-positive cells were obtained.

## Translating ribosome affinity purification

To purify ribosome-associated RNAs specifically from Purkinje neurons in the cerebellum from $Tsc1^{fl/fl}$ and $Tsc1^{+/+}$ animals, we used TRAP methods as previously described (*Heiman et al., 2014*; *Dougherty, 2017*). Briefly, 12 µl of Purified Recombinant Biotinylated Protein L (Pierce, cat #29997, 1 µg/µl) was conjugated to 30 µl of Streptavidin MyOne T1 Dynabeads (Thermo Fischer Scientific, cat # 65601) at room temperature using gentle end-over-end rotation. The beads were then washed five times with PBS containing 3% BSA (Jackson Immunoresearch, cat #001-000-162). After collecting beads on the magnet, they were washed once with low KCl wash buffer (10 mM HEPES pH 7.4, 5 mM $MgCl_2$, 150 mM KCl, and 1% NP-40) and further incubated with 100 µg total of mouse anti-GFP antibodies (HtzGFP-19F7 and HtzGFP-19C8, Memorial Sloan Kettering Centre, 50 µg of each) for 1 hr at room temperature.

To prepare cerebellar lysates, $Tsc1^{+/+}$ or $Tsc1^{fl/fl}$ L7Cre+/Rosa26fsTRAP+ animals were decapitated, and the cerebellum from each animal was homogenized in ice-cold homogenization buffer (10 mM HEPES pH 7.4, 5 mM $MgCl_2$, 150 mM KCl, 0.5 mM DTT, 100 µg/ml cycloheximide, RNasin, Superasin, and 1× protease inhibitors, cOmplete EDTA-free from Roche) using 12 strokes of Teflon-glass homogenizers. To clear the nuclei and debris, samples were centrifuged at 2000 × g for 10 mins at 4°C. Polysomes were prepared by adding NP-40 (1%) and DHPC (30 mM), incubating on ice for 5 min, followed by centrifugation at 20,000 × g for 15 min at 4°C. For input RNA, a 60 µl sample of the supernatant was removed, added to 190 µl of low KCl wash buffer and 750 µl of Trizol and stored at −80°C for RNA isolation. Remaining supernatant was incubated with GFP-coated Dynabeads overnight at 4°C with end-over-end rotation. After overnight incubation, anti-GFP beads were captured on the magnet, washed four times with high KCl wash buffer (10 mM HEPES pH 7.4, 5 mM $MgCl_2$, 350 mM KCl, 1% NP-40, 0.5 mM DTT, RnasinRNasin, and 100 µg/ml cycloheximide), and resuspended in 250 µl of low KCl wash buffer and 750 µl of Trizol reagent. RNA was isolated using Trizol reagent protocol. To remove contaminating genomic DNA, DNase I (Qiagen, cat #) was used from Qiagen RNAeasy Minelute kit. RNA quality was assessed by Bioanalyzer analysis using Pico chip. TRAP RNA from each animal was used to make Illumina ready libraries at the Molecular Biology Core facility at the Dana Farber Cancer Institute.

## RNA-Seq

Barcoded and pooled libraries were sequenced on Illumina HiSeq 4000 to generate 75 base paired end reads. Quality of RNA-seq data was assessed initially using FastQC (https://www.bioinformatics.babraham.ac.uk/projects/fastqc/), and Trimmomatic (http://www.usadellab.org/cms/?page=trimmomatic) was used to remove adapters and low quality bases. Reads were mapped to the mouse genome (Ensembl GRCm38) using STAR. For sorted PCs, the transcriptome was assembled with Cufflinks. For the TRAP data, the transcriptome assembly from the sorted PCs was used. Gene and transcript quantification were performed using Cufflinks. These data were imported into R for further analysis.

Differential expression from both datasets was calculated using the LIMMA package. Differentially expressed transcripts were also identified using LIMMA using the transcript-specific data from Cufflinks. Thresholds for differential expression were p-value<0.01 and fold change > 2.

Exon coverage was determined directly from the mapped SAM files for each sample using custom Python scripts. This was then converted into transcript coverage by concatenating exons into transcripts that were identified by Cufflinks. Transcripts were then broken into 100 evenly spaced intervals and average number of reads across an interval were normalized to the total number of reads across the entire transcript. For up- and downregulated transcripts, the sequences were obtained using the UCSC Table Browser based on coordinates from Cufflinks. Coding and non-coding (5'UTR and 3'UTR) were extracted from these sequences by searching for the longest open reading frame within the sequence. Coverage was then calculated separately for each portion of the transcript (5'UTR, CDS, 3'UTR), and the difference between wild-type and mutant cells was calculated. Gene ontology was performed using DAVID (https://david.ncifcrf.gov/), and non-redundant categories from GO_FAT Cellular Compartment, Biological Process, and Molecular Function were selected. Datasets of disease risk genes used for comparison included (1) ASD genes – SFARI (https://gene.sfari.org/) genes with a score of 1, (2) intellectual disability genes (*Parikshak et al., 2013*), and (3) schizophrenia risk genes (*Wang et al., 2018*).

TE was calculated as described in *Thoreen et al., 2012*. Briefly, counts from RNA and TRAP were transformed using $\log_2$(counts + 1), and $\Delta$TE = (TRAP $Tsc1^{fl/fl}$ – TRAP $Tsc1^{+/+}$) – (RNA $Tsc1^{fl/fl}$ – RNA $Tsc1^{+/+}$). Variance of the $\Delta$TE measure was estimated by recalculating $\Delta$TE 100 times per gene, while leaving one value out for each dataset. This distribution was summarized using a Z-score for each gene.

## Western blotting

$Tsc1^{fl/fl}$ or $Tsc1^{fl/fl}$ L7Cre animals (N = 6 for each genotype) were sacrificed at P42–45, and the cerebellum was dissected. The whole cerebellum was lysed in RIPA buffer with protease inhibitors (Pierce) and homogenized and sonicated. Lysates were clarified by centrifuging at 18,000 x g for 15 min at 4°C, and the insoluble pellets were discarded. The supernatant was quantified using a BCA assay. Laemmli buffer was added to the supernatant and boiled at 95°C for 5 min prior to loading onto 10% polyacrylamide gels, which were transferred onto PVDF membranes. Antibodies for western blots included DLG3 (PSD93) (Biolegend 818302) 1:1000, KIF3C (Proteintech 14333–1-AP) 1:800, CDK16 (PCTAIRE) (Proteintech 10102–1-AP) 1:750, RPL28 (Proteintech 16649–1-AP) 1:1000, GAPDH (Thermo Fisher AM4300) 1:10,000, and β-actin (Cell Signaling 3700) 1:5000. GAPDH was verified as a stable loading control by comparing it to the REVERT Total Protein stain (Licor) (*Figure 6—figure supplement 1*), and GAPDH was used for normalization for all other blots due to ease of quantification. Imaging was performed on a Licor fluorescent imager, and quantification was performed using ImageStudio (Licor).

## Immunohistochemistry

$Tsc1^{+/+}$ or $Tsc1^{fl/fl}$ L7Cre/L7GFP (N = 4 for each genotype) were perfused at P42 with saline and then 4% PFA. Brains were dissected, post-fixed in 4% PFA for 24 hr, and then placed in 30% sucrose for at least 24 hr. Floating sections (40 μm) were cut on a cryostat and stained with anti-GFP (Thermo Fisher A10262) and anti-SHANK2 (Thermo Fisher PA5-78652). Images were obtained on a spinning disk confocal microscope with 10 images per animal. Images were quantified using custom ImageJ macros. Briefly, GFP-positive dendrites were identified using a simple threshold, cell bodies were manually excluded, and this image was used as a mask to identify SHANK2-positive punctae. Intensity within each puncta was measured individually and averaged across the field of view.

## Statistical methods

Count data from RNA sequencing and TRAP were imported into R for further analysis. Differential expression from both datasets was calculated using the LIMMA package, and the thresholds for differential expression are indicated in the text. An ANOVA with Tukey's post hoc test was performed to find differences in coverage between sections of transcripts from up- and downregulated genes. . Other statistical tests were performed in R and are indicated in the text. For IHC and western blot

data, the average protein expression was imported into PRISM, and the comparison between wild type and mutant was performed using a T-test.

## Acknowledgements

We would like to thank Elizabeth Bainbridge and Sarika Gurnani for help with mouse breeding. The Sahin lab has received grant funding from the U.S. Army Medical Research Tuberous Sclerosis Complex Research Program (W81XWH-15-1-0189). Boston Children's Hospital Intellectual and Developmental Disabilities Research Center (BCH IDDRC, U54HD090255). JSD was supported by Roche Postdoctoral Fellowship (RPF) Program. KDW is funded by the Neuroscience Research Training Scholarship from the American Academy of Neurology and NIH (5K08NS112598). CLS is funded by the CH/BIDMC/Harvard Medical School Neurology Resident Research Education Program NIH R25NS070682.

## Additional information

### Competing interests

Meghan T Miller: Meghan T. Miller is affiliated with Roche Pharma Research and Early Development, Neuroscience and Rare Diseases and Skyhawk Therapeutics. The author has no financial interests to declare, MTM is now an employee of Skyhawk Therapeutics. Mustafa Sahin: Mustafa Sahin reports grant support from Novartis, Roche, Biogen, Astellas, Aeovian, Bridgebio, Aucta and Quadrant Biosciences. He has served on Scientific Advisory Boards for Novartis, Roche, Celgene, Regenxbio, Alkermes and Takeda. The other authors declare that no competing interests exist.

### Funding

| Funder | Grant reference number | Author |
| --- | --- | --- |
| American Academy of Neurology | Neuroscience Research Training Scholarship | Kellen Diamond Winden |
| National Institute of Neurological Disorders and Stroke | K08NS112598 | Kellen Diamond Winden |
| F. Hoffman-La Roche | Roche Postdoctoral Fellowship | Jasbir Singh Dalal |
| U.S. Army Medical Research Tuberous Sclerosis Complex Research Program | W81XWH-15-1-0189 | Mustafa Sahin |
| Eunice Kennedy Shriver National Institute of Child Health and Human Development | U54HD090255 | Mustafa Sahin |
| National Institute of Neurological Disorders and Stroke | R25NS070682 | Catherine Lourdes Salussolia |

The funders had no role in study design, data collection and interpretation, or the decision to submit the work for publication.

### Author contributions

Jasbir Singh Dalal, Conceptualization, Investigation, Writing - original draft, Writing - review and editing; Kellen Diamond Winden, Formal analysis, Investigation, Writing - original draft, Writing - review and editing; Catherine Lourdes Salussolia, Investigation, Writing - original draft, Writing - review and editing; Maria Sundberg, Achint Singh, Truc Thanh Pham, Investigation; Pingzhu Zhou, William T Pu, Resources; Meghan T Miller, Conceptualization, Resources, Supervision; Mustafa Sahin, Conceptualization, Supervision, Funding acquisition, Writing - original draft, Writing - review and editing

## Author ORCIDs
Kellen Diamond Winden (iD) https://orcid.org/0000-0001-5816-3724
Catherine Lourdes Salussolia (iD) http://orcid.org/0000-0003-0168-3205
Maria Sundberg (iD) http://orcid.org/0000-0002-6623-4411
Mustafa Sahin (iD) https://orcid.org/0000-0001-7044-2953

## Ethics
Animal experimentation: All animal procedures were carried out in accordance with the Guide for the Humane Use and Care of Laboratory Animals, and all procedures in this study were approved by the Animal Care and Use Committee of Boston Children's Hospital (Protocol number 18-05-3677R).

## Decision letter and Author response
Decision letter https://doi.org/10.7554/eLife.67399.sa1
Author response https://doi.org/10.7554/eLife.67399.sa2

## Additional files
### Supplementary files
- Source data 1. Raw and annotated western blot images.
- Supplementary file 1. Supplemental table 1.
- Supplementary file 2. Supplemental table 2.
- Supplementary file 3. Supplemental table 3.
- Supplementary file 4. Supplemental table 4.
- Supplementary file 5. Supplemental table 5.
- Transparent reporting form

### Data availability
Raw sequencing and processed data have been deposited in GEO DataSets under accession code GSE169719.

The following dataset was generated:

| Author(s) | Year | Dataset title | Dataset URL | Database and Identifier |
| --- | --- | --- | --- | --- |
| Dalal JS, Winden KD, Sahin M | 2021 | Loss of Tsc1 in cerebellar Purkinje cells induces transcriptional and translation changes in FMRP target transcripts | https://www.ncbi.nlm.nih.gov/geo/query/acc.cgi?acc=GSE169719 | NCBI Gene Expression Omnibus, GSE169719 |

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
