## [Decision Letter]

**Acceptance summary:**

This study reports that deletion of Tsc1 restricted to cerebellar Purkinje cells leads to reduced expression of FMRP target genes. Given the growing importance of mTOR signaling and cerebellar functions in ASD pathophysiology, the present work is an important and timely contribution to the field.

**Decision letter after peer review:**

Thank you for submitting your article "Loss of Tsc1 in cerebellar Purkinje cells induces transcriptional and translation changes in FMRP target transcripts" for consideration by *eLife*. Your article has been reviewed by 2 peer reviewers, and the evaluation has been overseen by a Reviewing Editor and Catherine Dulac as the Senior Editor. The following individuals involved in review of your submission have agreed to reveal their identity: Eric Klann (Reviewer #2).

Essential revisions:

1) The authors should examine the protein levels of at least one more FMRP target that has decreased RNA expression, but no change in ribosome-binding in the Tsc1 mutant PCs.

2) They should examine the protein levels of at least two of the 5'TOP mRNAs that have increased translation efficiency. In addition, they should reference previous findings by Das Sharma et al. (Cell Rep. 26: 3313) showing increased ribosome-bound 5'TOP mRNAs in fragile X mice.

3) Additional bioinformatic analysis should be done to determine whether any of the differentially regulated genes in either the RNA-seq or TRAP-seq data sets correspond to the length of the gene.

4) The key goal of the present study is to understand the molecular mechanisms underlying TSC1 deletion-dependent Purkinje cell dysfunctions using unbiased transcriptomic and translation analyses. Although Shank2 is certainly an important target of FRMP and has been shown to regulate key functions of Purkinje cells, it would have been much informative if the authors could have tested additional FMRP targets in their assays. Specifically, they analyze Shank2 protein levels in the cerebellum through IHC experiments. Here, they could have tried additional FMRP targets that are expressed in the cerebellum. In addition, immunoblot experiments could be performed additionally to test protein levels of Shank2 and other FMRP-target proteins, and also some negative (non-FMRP-target) controls.

5) The enrichment of the transcripts for FMRP targets is interesting. However, given that mTOR signaling is associated with various brain disorders, it would be nice if the authors could test enrichments for other gene sets (i.e. those associated with ASD and other brain disorders).

*Reviewer #1 (Recommendations for the authors):*

1. The key goal of the present study is to understand the molecular mechanisms underlying TSC1 deletion-dependent Purkinje cell dysfunctions using unbiased transcriptomic and translation analyses. Although Shank2 is certainly an important target of FRMP and has been shown to regulate key functions of Purkinje cells, it would have been much informative if the authors could have tested additional FMRP targets in their assays. Specifically, they analyze Shank2 protein levels in the cerebellum through IHC experiments. Here, they could have tried additional FMRP targets that are expressed in the cerebellum. In addition, immunoblot experiments could be performed additionally to test protein levels of Shank2 and other FMRP-target proteins, and also some negative (non-FMRP-target) controls.

2. The enrichment of the transcripts for FMRP targets is interesting. However, given that mTOR signaling is associated with various brain disorders, it would be nice if the authors could test enrichments for other gene sets (i.e. those associated with ASD and other brain disorders).

*Reviewer #2 (Recommendations for the authors):*

The authors should consider doing some additional validation/examination of their RNA-seq data as suggested above. First, they should examine the protein levels of at least one more FMRP target that has decreased RNA expression, but no change in ribosome-binding in the Tsc1 mutant PCs. Second, they should examine the protein levels of at least two of the 5'TOP mRNAs that have increased translation efficiency. In addition, they should reference previous findings by Das Sharma et al. (Cell Rep. 26: 3313) showing increased ribosome-bound 5'TOP mRNAs in fragile X mice. Finally, additional bioinformatic analysis should be done to determine whether any of the differentially regulated genes in either the RNA-seq or TRAP-seq data sets correspond to the length of the gene.

---

## [Author Response]

Essential revisions:1) The authors should examine the protein levels of at least one more FMRP target that has decreased RNA expression, but no change in ribosome-binding in the Tsc1 mutant PCs.

Based on this suggestion, we examined the protein expression of two other FMRP target genes, Dlg3 and Kif3c, and we found that these were both significantly down regulated in mutant cerebellum using western blots (revised Figure 6). We believe that these data strengthen our findings and would like to thank the reviewers for this important suggestion.

2) They should examine the protein levels of at least two of the 5'TOP mRNAs that have increased translation efficiency. In addition, they should reference previous findings by Das Sharma et al. (Cell Rep. 26: 3313) showing increased ribosome-bound 5'TOP mRNAs in fragile X mice.

As suggested, we investigated the protein levels of two predicted 5’TOP mRNAs that had increased translation efficiency, Cdk16 and Rpl28, but we did not observe any significant change in the protein levels of these genes. These data are included in the Figure 6—figure supplement 1. We have referenced the regulation of 5’ TOP genes by FMRP, and we suggest that the lack of an observed change in these genes may be due to contribution of non-Purkinje cells in the samples and/or other technical factors.

3) Additional bioinformatic analysis should be done to determine whether any of the differentially regulated genes in either the RNA-seq or TRAP-seq data sets correspond to the length of the gene.

We would like to thank the reviewers for this valuable suggestion. We have performed this analysis and found that genes with reduced expression in the 3’UTR show a significantly increased transcript length when compared both to all transcripts and to a random set of transcripts with a similar distribution of expression (Figure 2—figure supplement 1). As the reviewers suggested, these data are consistent with other observations of FMRP preferentially regulating longer transcripts, and we have added this point to the manuscript.

4) The key goal of the present study is to understand the molecular mechanisms underlying TSC1 deletion-dependent Purkinje cell dysfunctions using unbiased transcriptomic and translation analyses. Although Shank2 is certainly an important target of FRMP and has been shown to regulate key functions of Purkinje cells, it would have been much informative if the authors could have tested additional FMRP targets in their assays. Specifically, they analyze Shank2 protein levels in the cerebellum through IHC experiments. Here, they could have tried additional FMRP targets that are expressed in the cerebellum. In addition, immunoblot experiments could be performed additionally to test protein levels of Shank2 and other FMRP-target proteins, and also some negative (non-FMRP-target) controls.

As discussed above in essential revision 1, this is an important experiment to further explore our hypotheses. We have examined the expression of two addition FMRP targets, Dlg3 and Kif3c, and found that their expression was significantly reduced in mutant cerebellum. Unfortunately, we tried several times to analyze Shank2 expression using immunoblots but were unable to reliably detect the protein. As suggested by the reviewer, we have also examined the expression of some non-FMRP target genes and found them to be unchanged (Figure 6—figure supplement 1).

5) The enrichment of the transcripts for FMRP targets is interesting. However, given that mTOR signaling is associated with various brain disorders, it would be nice if the authors could test enrichments for other gene sets (i.e. those associated with ASD and other brain disorders).

As suggested, we have tested enrichment of genes with reduced 3’UTR expression against susceptibility genes for ASD, intellectual disability, and schizophrenia. Interestingly, we observed that these genes were enriched for ASD risk genes (SFARI genes with a score of 1), but we did not observe any enrichment of ID or schizophrenia genes. We have included these data in Figure 2—figure supplement 1.

Reviewer #1 (Recommendations for the authors):1. The key goal of the present study is to understand the molecular mechanisms underlying TSC1 deletion-dependent Purkinje cell dysfunctions using unbiased transcriptomic and translation analyses. Although Shank2 is certainly an important target of FRMP and has been shown to regulate key functions of Purkinje cells, it would have been much informative if the authors could have tested additional FMRP targets in their assays. Specifically, they analyze Shank2 protein levels in the cerebellum through IHC experiments. Here, they could have tried additional FMRP targets that are expressed in the cerebellum. In addition, immunoblot experiments could be performed additionally to test protein levels of Shank2 and other FMRP-target proteins, and also some negative (non-FMRP-target) controls.

We would like to thank this reviewer for their comments and suggestions. We have added data based on this suggestion, which is described in essential revisions #4 above.

2. The enrichment of the transcripts for FMRP targets is interesting. However, given that mTOR signaling is associated with various brain disorders, it would be nice if the authors could test enrichments for other gene sets (i.e. those associated with ASD and other brain disorders).

We have added these data to the manuscript, and the details are in essential revisions #5 above.

Reviewer #2 (Recommendations for the authors):The authors should consider doing some additional validation/examination of their RNA-seq data as suggested above. First, they should examine the protein levels of at least one more FMRP target that has decreased RNA expression, but no change in ribosome-binding in the Tsc1 mutant PCs. Second, they should examine the protein levels of at least two of the 5'TOP mRNAs that have increased translation efficiency. In addition, they should reference previous findings by Das Sharma et al. (Cell Rep. 26: 3313) showing increased ribosome-bound 5'TOP mRNAs in fragile X mice. Finally, additional bioinformatic analysis should be done to determine whether any of the differentially regulated genes in either the RNA-seq or TRAP-seq data sets correspond to the length of the gene.

We would like to thank the reviewer for their attentive examination of our manuscripts and their helpful comments. We have responded to these suggestions in essential revisions #2 and #3.